# YAP/TAZ Signalling in Colorectal Cancer: Lessons from Consensus Molecular Subtypes

**DOI:** 10.3390/cancers12113160

**Published:** 2020-10-28

**Authors:** Sophie Mouillet-Richard, Pierre Laurent-Puig

**Affiliations:** 1Centre de Recherche des Cordeliers, INSERM, Sorbonne Université, Université de Paris, F-75006 Paris, France; pierre.laurent-puig@parisdescartes.fr; 2Institut du Cancer Paris CARPEM, APHP, Department of Biology, Hôpital Européen Georges Pompidou, F-75015 Paris, France

**Keywords:** YAP/TAZ, colorectal cancer, molecular classification, prion protein

## Abstract

**Simple Summary:**

Colorectal cancer (CRC) is a heterogeneous disease that can be divided into 4 consensus molecular subtypes (CMS) according to molecular profiling. The CMS classification is now considered as a reference framework for understanding the heterogeneity of CRC and for the implementation of precision medicine. Although the contribution of YAP/TAZ signalling to CRC has been intensively studied, there is little information on its role within each CMS subtype. This article aims to provide an overview of our knowledge of YAP/TAZ in CRC through the lens of the CMS classification.

**Abstract:**

Recent advance in the characterization of the heterogeneity of colorectal cancer has led to the definition of a consensus molecular classification within four CMS subgroups, each associated with specific molecular and clinical features. Investigating the signalling pathways that drive colorectal cancer progression in relation to the CMS classification may help design therapeutic strategies tailored for each CMS subtype. The two main effectors of the Hippo pathway YAP and its paralogue TAZ have been intensively scrutinized for their contribution to colon carcinogenesis. Here, we review the knowledge of YAP/TAZ implication in colorectal cancer from the perspective of the CMS framework. We identify gaps in our current understanding and delineate research avenues for future work.

## 1. Introduction

Colorectal cancer (CRC) represents the third most frequent cancer with over 1,800,000 cases and the second cause of cancer-related death with over 880,000 deaths worldwide in 2018 according to the World Health Organization (Globocan 2018). Over the last decade, it has become apparent that CRC is a heterogeneous disease, and that the diversity of CRC encompasses multiple parameters: anatomical location (distal/proximal), anatomopathological stages (so-called stages I to IV), genomic hallmarks (chromosomal instability, microsatellite instability, hypermethylator phenotype) [1]. More recently, large scale transcriptomic analyses have allowed one to uncover the molecular heterogeneity of CRC and have led to the definition of a molecular classification into 4 consensus molecular subtypes (CMS) [2]. The overall characteristics of each CMS subgroup are summarized in Figure 1.

CMS1 tumours (14% of patients) are frequently associated with microsatellite instability as well as an hypermethylator phenotype and mutations within the BRAF gene. CMS2 (37% of patients) is the so-called canonical subtype. CMS2 tumours are often associated with chromosomal instability, copy number alterations and activation of the WNT/MYC pathway. The CMS3 subtype (13% of patients), also termed “metabolic”, is enriched in *KRAS* mutations. Lastly, the CMS4 subtype (23% of patients) is characterized by an epithelial to mesenchymal (EMT) signature as well as activation of the TGFß pathway. Other hallmarks of this “mesenchymal” subtype include matrix remodelling, wound response, angiogenesis and cancer stem cell (CSC) signatures. Each subtype is further associated with a specific tumour microenvironment, such as immune infiltration for CMS1 tumours and stromal infiltration in the case of CMS4 tumours [3]. Finally, the CMS classification has a strong prognostic value. Indeed, CMS1 patients have a better overall survival (OS) but a worse survival after relapse. On another hand, the CMS4 subgroup is associated with unfavourable prognosis in terms of OS and relapse-free survival [2]. The CMS4 subtype is more prevalent in advanced stages of CRC and is also associated with increased progression rates to more advanced stages [4]. The CMS classification is now considered the gold standard to depict the molecular heterogeneity of CRC and constitutes a path towards the development of precision medicine [5]. Indeed, several authors have recommended taking into account this classification when analysing or designing clinical trials [5,6]. Importantly, the CMS classification may also have major implications with respect to the pathways orchestrating tumour growth and response to therapy.

The Hippo pathway, initially discovered in Drosophila in the late 1990s, is highly conserved from fly to mammals (reviewed in [7]). Over the years, it has gained tremendous interest as a major regulator of cell growth and organ size. The Hippo cascade has been described in detail in several reviews [8,9,10,11]. Schematically, the core components include the kinases MST1/2 (the mammalian homologues of Drosophila Hippo) and LATS1/2 acting upstream of the effector YAP (Yes-associated protein) and its paralogue TAZ (transcriptional co-activator with PDZ-binding motif). Activation of the pathway promotes phosphorylation and inactivation of the two transcriptional coactivators YAP and TAZ. To the opposite, the Hippo pathway inactivation triggers the nuclear translocation of YAP/TAZ, where they can associate with different transcription factors to regulate gene transcription (Figure 2).

In line with their contribution to cell survival and proliferation, YAP and TAZ have been shown to play essential roles in several key aspects of cancer initiation, progression and metastasis (reviewed in [12]). In the present review, we aim to revisit the literature relating to YAP/TAZ in CRC through the lens of molecular classification. We provide an overview of clinical, animal, cellular and molecular data on YAP/TAZ in the context of CRC and discuss their implication in the framework of the CMS subtyping of CRC.

## 2. YAP/TAZ in Colorectal Cancer: From Cells to Patients

### 2.1. Clinical Data

At the genomic level, *YAP1* (encoding YAP) or *WWTR1* (encoding TAZ) genetic alterations appear to be extremely rare in CRC (less than 2% in 4541 CRC samples according to cBioportal http://www.cbioportal.org/ [13,14]). However, a single nucleotide polymorphism (SNP) located within the *YAP1* gene (rs2186607) was identified as a common genetic risk variant for sporadic CRC in a very recent meta-analysis covering over 125,000 individuals, with a Hazard Ratio of 1.05 [15].

At the transcriptional level, Yuen et al. exploited two large CRC cohorts to assess the prognostic impact of YAP and TAZ expression as well as that of their two target genes *AXL* and *CTGF* [16]. Expression of the TAZ-encoding *WWTR1* gene was highly correlated with that of *AXL* and *CTGF* and found to be associated with shorter survival, while this was not observed for YAP. In these cohorts, the combined overexpression of *WWTR1*, *AXL* and *CTGF* was particularly unfavourable [16]. These conclusions were confirmed in a further study [17], which leveraged the large CRC dataset contributed by our group [18]. A study based on a collection of liver metastases resected from CRC patients having received or not having received neo-adjuvant chemotherapy further established that *YAP1* mRNA levels were increased in treated vs. untreated patients and that *YAP1* expression correlated with poor OS and disease-free survival [19]. In line with this, Lee and colleagues extracted a *YAP1* gene activation signature from CRC cell microarray data and documented that the activated *YAP1* signature was associated with poor prognosis across four stages I to III CRC cohorts, including the two cohorts analysed in the Yuen study [20]. Furthermore, in patients with metastatic disease and wild-type *KRAS*, this gene signature of YAP activation was found to be associated with resistance to cetuximab [20], a monoclonal antibody against the EGFR receptor recommended as the first-line treatment in such patients in combination with a FOLFOX (5-Fluorouracil or 5-FU and oxaliplatin) or FOLFIRI (5-FU and irinotecan) regimen [21].

Finally, at the protein level, Steinhardt and colleagues provided the first immuno-histochemical data demonstrating that YAP is overexpressed in a panel of 28 CRC specimens [22]. These observations were subsequently corroborated on two larger studies including 168 [23] and 139 [24] CRC cases, respectively. Interestingly, patients exhibiting higher staining of both YAP and TAZ in the former study were found to have shorter survival times [23]. The association between high YAP expression and shorter survival was confirmed in the later study, which also found a positive correlation with Tumor (T), Nodes (N), Metastases (M) stage and CyclinD1 overexpression [24]. To the opposite, a study evaluating the expression of YAP through immunochemistry in a large cohort of 672 CRC patients concluded that the complete loss of YAP staining, occurring in a minor subset of the tumours, was associated with worse survival [25].

Finally, considering CRC through the prism of the CMS classification, while transcripts encoding YAP are slightly overrepresented in both CMS2 and CMS4 samples, there is a clear-cut enrichment in transcripts encoding TAZ in CMS4 tumours (Figure 3).

Moreover, CMS4 tumours exhibit enrichment of the YAP/TAZ target score, which corresponds to the integrated mRNA expression of a selection of 22 YAP/TAZ target genes and was shown to represent a robust index of YAP/TAZ activation [26] (Figure 3). Finally, at the protein level, mining The Cancer Proteome Atlas [27] reveals that YAP and TAZ protein levels are enriched in CMS4 tumours (Figure 3).

Overall, the association between high YAP/TAZ expression and poor prognosis fully fits in with their overrepresentation in the CMS4 subgroup, which is that of worse prognosis.

### 2.2. Insight from Animal Models

Contrary to the clinical data globally supporting an unfavourable association between high YAP/TAZ expression and patient outcome, the issue as to whether YAP and/or TAZ activation may promote or suppress CRC based on animal studies remains highly controversial. The first set of studies indicates that Yap over-activation promotes dysplastic growth of the mouse intestine with an expansion of undifferentiated progenitor cells [28,29]. From an experimental point of view, two different approaches were used: over-expression of a constitutively active form of Yap, Yap^S127A^, under the control of doxycycline [28], or constitutive knockout of Mst1 combined with conditional deletion of Mst2 in the intestine (association of an Mst2^fl/fl^ construction with the expression of the Cre recombinase under the control of the Villin promoter, i.e., Mst1^-/-^Mst2^fl/fl^/Villin-Cre mice) [29]. The observations reported in these two seminal works are in full agreement with the indispensable role of Yap in the regeneration of intestinal stem cells (ISCs), conserved from Drosophila to mice (reviewed in (Yu et al., 2015)), and the notion that CRC arises from uncontrolled expansion of ISCs [30]. Interestingly, Yap activation was shown to be a direct consequence of Apc loss of function in the Apc^Min^ mouse model of colon cancer [31]. Inactivation of APC (Adenomatous Polyposis Coli) represents the most frequent early event in colorectal tumorigenesis and the inactivation of Apc in intestinal cells is a well-established approach to recapitulate colon cancer in mice [32]. Cai et al. further found that YAP activation is a general hallmark of tubular adenomas of patients with Familial Adenomatous Polyposis (FAP), an inherited disease associated with somatic mutations in the APC gene. At a molecular level, Apc, which behaves as a molecular scaffold and is best known for its negative regulation of the canonical Wnt pathway [33], was shown to operate upstream of Lats1 activation [31]. Importantly, conditional inactivation of either Yap or Taz in the intestine (Yap^fl/fl^ or Taz^fl/fl^ constructions associated with Villin-Cre) abolished adenoma formation in Apc^Min^ mice [31]. Consistent with these studies, Gregorieff and colleagues demonstrated that inducible Yap inactivation (Yap^fl/fl^ associated with a Villin-CreERT2 construct allowing for a tamoxifen-inducible deletion of Yap) impairs recovery of the intestinal epithelium after exposure to ionizing radiation and abolishes adenomas in Apc^Min^ mice [34]. However, the authors found no effect of Yap and/or Taz conditional deletion on crypt hyperplasia induced by acute loss of Apc (Apc^fl/fl^ with VillinCreERT2), contrary to the report by Azzolin et al. [35]. In a follow-up study based on single-cell analysis, the Gregorieff-Wrana group depicted a critical role for Yap in the expansion of a unique stem cell compartment designated revival stem cells (revSCs) upon various types of intestinal damage: irradiation, targeted ablation of Lgr5^+^ ISCs residing at the crypt base or treatment with dextran sodium sulfate (DSS) [36]. Interestingly, these revSCs were shown to be endowed with the capacity to reconstitute the Lgr5^+^ crypt base compartment and to regenerate a functional intestine. Whether these revSCs can support tumour growth remains to be investigated. Another issue that would be worth addressing is their resemblance with Sca1^+^ reserve-like stem cells, a cell population found to express a strong regenerative/tumorigenic potential driven by Yap, which can be turned on in response to prostaglandin signals emanating from the mesenchymal niche [37].

These overall observations are however challenged by a series of studies arguing that YAP instead plays a tumour suppressive role in CRC. Thus, Barry et al. found that inducible over-activation of Yap in intestinal cells interfered with the renewal of ISCs and that conditional loss of Yap in the intestine promotes crypt hyperplasia after whole-body irradiation [25]. Accordingly, the authors reported that DLD1 CRC cells engineered to overexpress wild-type YAP or a constitutive YAP mutant (YAP^S127A^) induced smaller tumours than control cells in xenograft experiments. Finally, Cheung et al. exploited multiple transgenic models and intestinal damage paradigms with the view to resolve the YAP CRC-promoting versus -suppressing conundrum [38]. Their results obtained upon conditional over-activation of Yap indicate a restriction of Lgr5^+^-derived cells, suggesting that Yap interferes with the self-renewal of Lgr5^+^ ISCs. Conversely, conditional inactivation of Yap and Taz was associated with a higher tumour burden in a model of acute loss of Apc or in the carcinogen-induced CRC model of azoxymethane injection [38].

Thus, the debate over the pro- versus the anti-tumorigenic role of YAP/TAZ in CRC is clearly ongoing. The discrepancy between the various studies may be accounted for, at least partly, by the different transgenic models used (e.g., Apc^Min^ mice versus VillinCreERT2-Apc^fl/fl^ mice) or the different time points for analysis. Also, as underlined by [11], the results may be influenced by the cell type(s) where YAP/TAZ are manipulated (whole intestine versus a specific stem cell compartment for instance) as well as the resulting sub-cellular location of YAP/TAZ potentially yielding opposite control on Wnt activity (inhibition of ß-catenin when YAP/TAZ are cytosolic, potentiation of ß-catenin activity when YAP/TAZ are nuclear).

Another approach to potentially accommodate the divergent conclusions is to reason in terms of CMS. Getting back to the proteomic data of The Cancer Proteome Atlas [27], both YAP and TAZ protein levels are increased in CMS4 tumours, while they are decreased in CMS2 and CMS3 tumours (Figure 3). Thus, we may envision that reduced YAP/TAZ levels favour the progression of CMS2 and CMS3 subtypes of CRC while their over-activation sustains that of CMS4 tumours. In this respect, it is worth noting that the different CMS may arise from distinct cellular origins and that the cell-of-origin influences the wiring of signaling pathways sustaining cell transformation [39]. Incidentally, most mouse models of CRC used in the studies cited above mimic the canonical subgroup CMS2 [40]. In addition, animal models recapitulating the features of CMS4 tumours were lacking [40], until the recent development of a transgenic model combining Notch1 over-activation with conditional Tp53 deletion and Kras mutation (“KPN” model) [41]. Interestingly, two of the studies supporting the Yap pro-tumorigenic hypothesis had depicted an activation of the Notch pathway downstream from Yap [28,29]. Indeed, mice with an over-activated isoform of Yap were found to express increased levels of the Notch target gene *Hes1*, and treating these mice with a Notch inhibitor alleviated Yap-induced dysplasia [28]. Similarly, Zhou et al. reported an upregulation of Hes1 in the intestine of Mst1^-/-^Mst2^fl/fl^/Villin-Cre mice, together with an increase in the mRNA and protein levels of the Notch ligand Jagged1 [29]. Incidentally, JAGGED1 was subsequently identified as a YAP target gene [42]. Thus, as YAP/TAZ appears to be upstream of NOTCH, investigating the Hippo pathway in the KPN model may lack relevance. However, whether Yap and/or Taz over-activation in a Tp53-deleted Kras-mutated context recapitulates the KPN model would be worth investigating. An alternative approach would be to examine the impact of YAP/TAZ over-activation or deletion in a panel of cell lines representative of each CMS on their tumorigenic potential [43], as reported by Barry et al. for the DLD1 cell line [25], which unfortunately could not be ascribed a CMS label [43]. In this respect, we may note that the stable knockdown of YAP in the SW620 cell line, predicted as CMS4 [43], is associated with a reduced tumorigenic potential in mice [44].

Obviously, resolving the question as to whether YAP/TAZ promotes or restricts cancer progression according to the CRC subtype considered is a prerequisite for the potential development of YAP/TAZ targeted therapeutic strategies [45].

### 2.3. The Cellular Scale

Convergent data have shown the expression of abundant YAP levels in various lines of CRC [23,29,44]. A recurrent observation is that the knockdown of YAP and/or TAZ inhibits the proliferation of CRC cells [23,46,47,48], their growth in soft agar [16,29,44], migration [49], invasion [50], CSC-like attributes [51,52] or tumorigenic properties in xenograft experiments [16,44,47,53,54]. We may note, however, that the tumorigenic role of YAP/TAZ in cell-grafted mice was challenged by opposite results obtained with DLD1 cells by Barry and colleagues [25]. Two studies from the Hahn’s group further depicted an essential role of YAP for the survival and tumorigenesis of various ß-catenin-driven CRC cell lines [53] or *KRAS*-mutated CRC cell lines submitted to KRAS suppression [55]. The later study notably described a YAP-dependent transcriptional programme of EMT [55]. Similarly, YAP depletion enhances the efficacy of BRAF or MEK inhibitors both in vitro and in xenografted mice [56]. Along the same line, YAP and/or TAZ were shown to mediate the tumour growth-supporting effect of endothelin-1 [54], REGγ, a promoter of bowel inflammation [48], or ISLR, a stromal protein involved in intestinal epithelial regeneration [57]. YAP contributes to the induction of EMT markers and migratory and invasive properties ensuing from the downregulation of the tumour suppressors RARγ [58] or HACE1 [59] or in an established metastatic derivative of the HCT116 cell line [60]. In addition, YAP endows CRC cells with the ability to actively migrate within the vasculature and thereby enhance their metastatic spread [61]. Consistent evidence further indicates that YAP is required downstream from HIF2α [62] or GPRC5A [63] for CRC cell growth under hypoxia. A different aspect of YAP function in colon cancer cells is its requirement for the maintenance of stem cell markers, which involves the negative regulation of the CDX2 transcription factor [64]. Finally, Touil et al. have documented increased YAP protein levels in response to 5-FU or oxaliplatin, which has to be brought together with the increased levels of *YAP1* mRNA in liver metastases from chemotherapy-treated versus untreated CRC patients [19]. Along the same line, an elevation of YAP expression and activity was found in 5-FU resistant CRC cells [65,66]. Altogether, cell-based approaches have yielded extensive support for the contribution of YAP/TAZ to many aspects of CRC. Some of those, such as EMT, resistance to 5-FU, and CSC hallmarks represent typical features of CMS4 CRC [2,67].

Accordingly, when combining transcriptomic data from [68] and molecular classification of cell lines according to [43], we observe that mRNAs encoding YAP and TAZ are elevated in CRC cell lines belonging to the CMS4 subtype, although CMS1 cell lines also express high levels of transcripts encoding TAZ (Figure 4). In addition, as observed with clinical samples, CMS4 CRC cell lines display much higher levels of the YAP/TAZ target score (Figure 4), a pan-cancer signature of YAP/TAZ activation [26]. In agreement, YAP and TAZ protein levels, as determined in [69], are also significantly more abundant when comparing CMS4 cells to all other subgroups (Figure 4).

In line with the above observations, we recently documented that YAP and/or TAZ control a panel of genes that specify the CMS4 phenotype [70]. Indeed, the knock-down of YAP and/or TAZ in the CMS4-type MDST8 colon cancer cell line causes a reduction in expression of the *AXL*, *CDH2* or *ZEB1* genes [70], all associated with EMT [71,72]. Our unpublished data also indicate a decrease in the expression of the CSC marker *CD44* upon YAP/TAZ silencing. As a whole, similarly to the TGFß pathway [2], YAP/TAZ should now be recognized as major determinants of the CMS4 subtype of CRC.

## 3. Upstream Pathways

From a transcriptional point of view, several studies employing colon cancer cells have depicted a regulation of YAP1 transcription by ß-catenin [44], HIF1α [50], p65 NFκB [48], TEAD4 [73] or CREB [74] (Figure 5). As for TAZ, it may be induced by YAP itself, and possibly by ß-catenin [31]. Besides, several miRNA-dependent mechanisms of YAP and TAZ regulation in tumour progression have been described [75]. In this respect, it is worth noting that TAZ is a direct target of miR-200a in CRC cells [26] and that miRNAs of the miR-200 family are reduced in CMS4 CRC [2].

At the translational level, beyond a classical regulation downstream from the MST1/2 or LATS1/2 kinases [31,48,63], Azzolin et al. described a mechanism whereby TAZ protein levels are directly controlled by the APC-containing ß-catenin destruction complex [46] (Figure 6). Another non-canonical pathway for regulation of YAP activity documented by the group of M. Karin is the Gp130-Src family kinase (SFK) axis, which provides a potential mechanism for inflammation-driven, Hippo-independent YAP activation in CRC [76,77]. Finally, other well-described upstream regulators of YAP/TAZ include various types of mechanical inputs (see [9,78,79] for review). Although not examined to our knowledge in the context of CRC, the mechanotransduction-dependent activation of YAP/TAZ was found to play an important role in intestinal repair after tissue damage [80].

In the context of CMS4 CRC, we recently established that YAP/TAZ are under the control of the cellular prion protein, denoted PrP^C^, a protein mainly known for its involvement in neurodegenerative disease [81] and whose role in cancer progression is attracting interest [82,83] (see Figure 6). More precisely, we uncovered that expression of the PrP^C^-encoding gene *PRNP*, which is enriched in CMS4 tumours, is associated with a YAP/TAZ signature in CRC patients and cell lines [70]. Going one step further, we found that depleting MDST8 CMS4 cells from PrP^C^ resulted in increased phospho-YAP to YAP ratio, decreased TAZ protein levels and reduced expression of several YAP/TAZ canonical target genes [70]. As PrP^C^ is a plasma membrane-anchored protein and YAP/TAZ are cytoplasmic or nuclear, delineating the molecular cascade linking PrP^C^ to YAP/TAZ obviously requires further investigation. Intriguingly, Besnier et al. reported a direct interaction of PrP^C^ with YAP in both the cytoplasm and nucleus of several CRC cell lines [84]. Albeit poorly investigated, the nuclear targeting of PrP^C^ indeed appears to occur under specific conditions such as genotoxic stress for instance [85]. In this respect, it is worth noting that CMS4 tumours display downregulation of all DNA repair pathways, which is proposed to result from hypoxia-induced oxidative stress [86]. Thus, whether the interplay between PrP^C^ and YAP/TAZ in CMS4 CRC also involves direct interaction and is sustained by genotoxic or oxidative stress conditions is worth considering.

Besides, another candidate upstream regulatory pathway of YAP/TAZ activation in CMS4 CRC is the VEGF (vascular endothelial growth factor)-VEGFR axis (Figure 6). Indeed, VEGF-VEGFR signaling is overrepresented in CMS4 patients [2] or cell lines [67] and has been reported to activate YAP/TAZ in both tumour cells and endothelial cells [87]. Thus, beyond a contribution of VEGFR-dependent YAP/TAZ activation in endothelial cells to tumour angiogenesis, an attractive hypothesis is that YAP/TAZ activation downstream from VEGFR in tumour cells themselves would foster vasculogenic mimicry, a process whereby tumour cells promote the formation of microvascular channels to supply blood into the tumour independently from endothelial cells [88].

Finally, because CMS4 tumours are characterized by high stromal, most notably fibroblast, infiltration [3], it is tempting to speculate that YAP/TAZ activation in CMS4 tumours cells may also be, at least partly, driven by signals emanating from the tumour microenvironment such as PGE2 [37,74] (see Figure 6). Other potential upstream regulators that are worth investigating are matrix stiffness [89], which may be fostered by surrounding cancer-associated fibroblasts [90], or hypoxia [62,63], which is overrepresented in CMS4 tumours [86]. Based on the various potential upstream activators of YAP/TAZ in CMS4 CRC, we may further surmise that if YAP/TAZ are activated in some way and co-operate with the adequate partner, colon cancers gain CMS4 properties and that the high activation of YAP/TAZ is essential to maintain CMS4 phenotype.

## 4. Choosing the Adequate Partner

Contrary to the main developmental pathways (Notch, Hedgehog, Wnt, TGFß/BMP) which comprise dedicated transcription factors, the Hippo cascade does not operate on its own but rather intersects with other pathways. Indeed, effectors YAP and TAZ cannot bind DNA directly but instead act as transcriptional co-regulators [11]. Thus, one difficulty in defining the transcriptional programme regulated by YAP/TAZ is that it may greatly vary according to the partner(s) involved. This response may also depend on the activation status of other major signalling cascades since multiple cross talks have been uncovered between YAP/TAZ and the Notch, Hedgehog, Wnt and TGFß/BMP pathways (see [11,91,92] for review).

A consensus view is that YAP/TAZ binds to DNA through their association with the TEA domain family members TEAD1-4 [9,78] (Figure 7). Interestingly, the activity of YAP/TAZ may be regulated by the palmitoylation of TEAD factors [93], or their nuclear availability, itself controlled by p38-dependent phosphorylation [94]. Furthermore, the association of YAP/TAZ with TEAD can be counteracted by VGLL4, which not only competes for TEAD binding but also disrupts the TEAD4-TCF4 complex and thereby negatively regulates both YAP- and ß-catenin-dependent signalling [95]. Another negative regulator of YAP/TAZ in CRC is TIAM1 whose action is twofold: it both enhances the degradation of cytoplasmic TAZ and suppresses the interaction between YAP/TAZ and TEAD [96]. Besides, the transcriptional activity of YAP/TAZ can be modulated by a plethora of other factors, including transcription factors such as AP-1, SMAD, RUNX2 or p73 or chromatin remodelling complexes such as NuRD or SWI/SNF [9,78].

More specifically, in the context of intestinal tissue, associations with AP-1 [55,97], ß-catenin and TBX5 [53], PKM2 [98], KLF4 [99] or KLF5 [52] have been reported (Figure 7).

Interestingly, the outcome of YAP/TAZ activity may vary according to the associated partner. Indeed, Imajo et al. found that YAP/TAZ sustain ISC proliferation when combined with TEAD while promoting goblet cell differentiation when complexed with KLF4 [99]. A still unresolved question is whether the choice of partner is dictated by post-translational modifications, similarly to the SEDT7-dependent methylation of YAP shown to modulate the interplay with ß-catenin [100]. Another issue that would be worth addressing is a potential concerted action with the ZEB1 transcription factor, as described in the context of breast cancer [101,102] as ZEB1 represents one of the CMS4-classifier genes [43]. Likewise, probing cooperation of YAP/TAZ with TGFß-activated SMADs, described in several contexts [91,92,103], seems relevant given the well-established contribution of the TGFß pathway in CMS4 CRC [2]. Such a model is supported by the combined control by YAP/TAZ and TGFß on a panel of CMS4 genes downstream from PrP^C^ [70]. Finally, a provocative hypothesis would be the modulation of YAP/TAZ activity through their binding with PrP^C^, in line with the observation of YAP/PrP^C^ complexes [84], and the reports that nuclear PrP^C^ binds chromatin [104,105].

As a whole, considering YAP/TAZ by themselves appears as an over-simplistic view of their activity and downstream action. Hence, further investigating YAP/TAZ through the lens of the CMS classification necessitates an extensive study of their potential binding partners in relation to CMS subtypes, and most notably in CMS4 tumors. We believe such studies would help shed light on the issue as to whether YAP/TAZ are drivers of CMS4 CRC, as described for TGFß.

## 5. Conclusions/Future Prospects

While studies focusing on the contribution of YAP/TAZ to CRC have yielded much insight over the past years, the investigation of YAP/TAZ within the framework of the CMS taxonomy of CRC is only in its infancy. We suggest that a thorough exploration of YAP/TAZ binding partners, upstream signals and regulators, target genes and biological action according to CMS subtypes would greatly improve our understanding of their accurate role. Although scarce, a body of evidence, summarized in this review, supports the notion that YAP/TAZ plays an important role in CMS4 CRC. Reaching an integrated understanding of the biological contribution of YAP/TAZ to the emergence or maintenance of CMS4 tumours helps design new therapeutic strategies towards this poor-prognosis subtype of CRC.

## Figures and Tables

**Figure 1 cancers-12-03160-f001:**
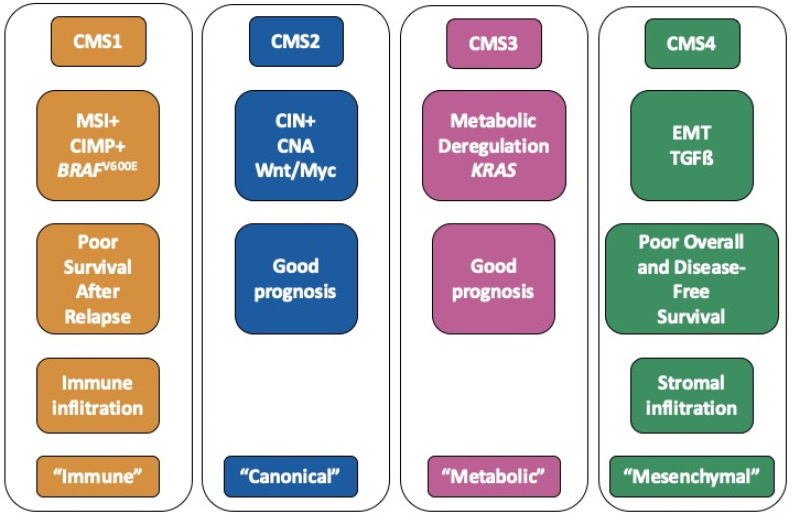
Summary of the hallmarks of each consensus molecular subtypes (CMS) subgroup.

**Figure 2 cancers-12-03160-f002:**
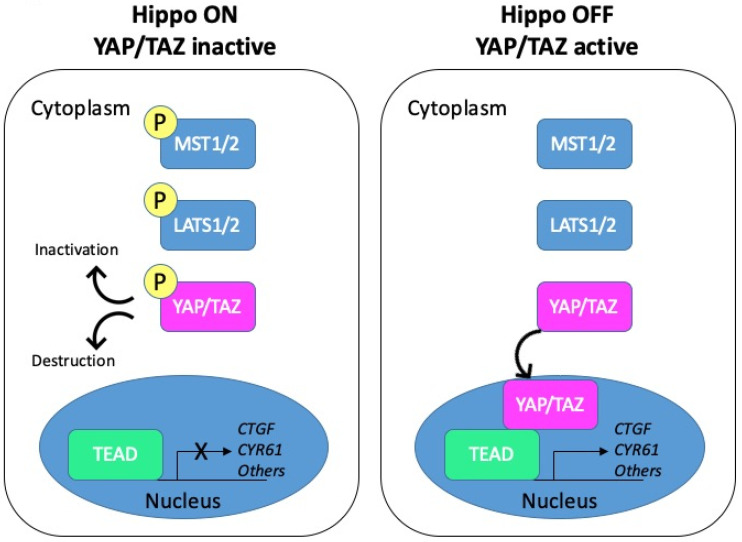
Schematic representation of the Hippo pathway. Left panel: when the Hippo pathway is on, the core kinases MST1/2 are phosphorylated and in turn phosphorylate the LATS1/2 kinases. Activated LATS1/2 phosphorylate YAP/TAZ, which promotes their cytoplasmic retention through binding to 14-3-3 proteins or their degradation. YAP/TAZ are thus inactive. Right panel: when the Hippo pathway is off, YAP/TAZ are dephosphorylated and can shuttle to the nucleus. Active YAP/TAZ bind transcriptional factors and induce the transcription of target genes such as *CTGF* or *CYR61*.

**Figure 3 cancers-12-03160-f003:**
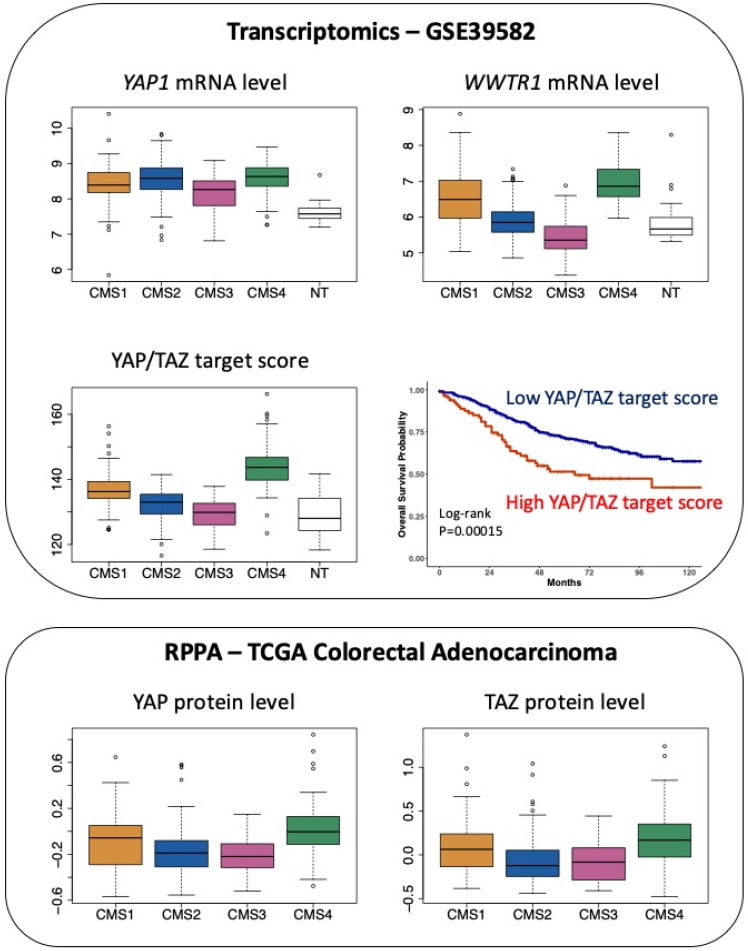
Expression of YAP and TAZ in CRC patients according to the CMS classification. Top panel: expression of *YAP1* mRNA and *WWTR1* (encoding TAZ) mRNA as well as the determination of the YAP/TAZ target score as defined in [26] in the 566 CRC patients of the GSE39582 dataset [18] according to the CMS classification [2]. *p* < 0.05 for CMS4 versus CMS1 and *p* < 0.0001 for CMS4 versus CMS3 or versus NT respectively for *YAP1* (Tukey post hoc test). *p* < 0.0001 for CMS4 vs. other CMS or vs. NT for *WWTR1* and YAP/TAZ target score. NT indicates non-tumour samples. Kaplan–Meier overall survival according to high and low YAP/TAZ target score was determined in the GSE39582 dataset. Bottom panel: protein levels of YAP and TAZ determined through Reverse Phase Protein Array (RPPA) in The Cancer Proteome Atlas of colorectal adenocarcinoma [27], according to the CMS classification [2]. *p* < 0.05 for CMS4 vs. CMS1 and *p* < 0.0001 for CMS4 vs. CMS2 or vs. CMS3 for YAP. *p* < 0.0001 for CMS4 vs. CMS2 or vs. CMS3 for TAZ (Tukey post hoc test).

**Figure 4 cancers-12-03160-f004:**
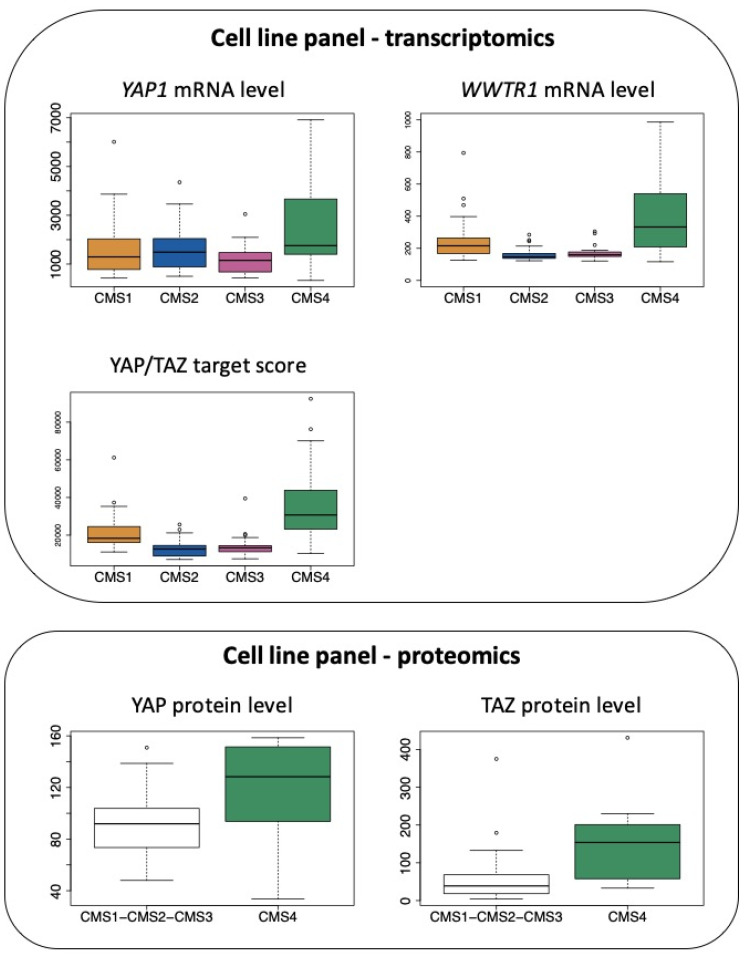
Expression of YAP and TAZ in CRC cell lines according to the CMS classification. Top panel: expression of *YAP1* mRNA and *WWTR1* mRNA as well as the determination of the YAP/TAZ target score as defined in [26] in the CRC cell line panel of the GSE59857 dataset [68] according to the CMS classification reported in [43]. *p* < 0.05 for CMS4 vs. CMS1 and *p* < 0.01 for CMS4 vs. CMS2 or vs. CMS3 for *YAP1*. *p* < 0.001 and *p* < 0.005 for CMS4 vs. CMS2 or vs. CMS3, respectively, for *WWTR1*. *p* < 0.0001 for CMS4 vs. other CMS for YAP/TAZ target score (Tukey post hoc test). Bottom panel: protein levels of YAP and TAZ determined in the CRC cell line panel proteomic study [69], in CMS4 cell lines vs. other cell lines. *p* < 0.05 for CMS4 vs. other CMS for YAP and TAZ (Kruskal–Wallis chi-squared test).

**Figure 5 cancers-12-03160-f005:**
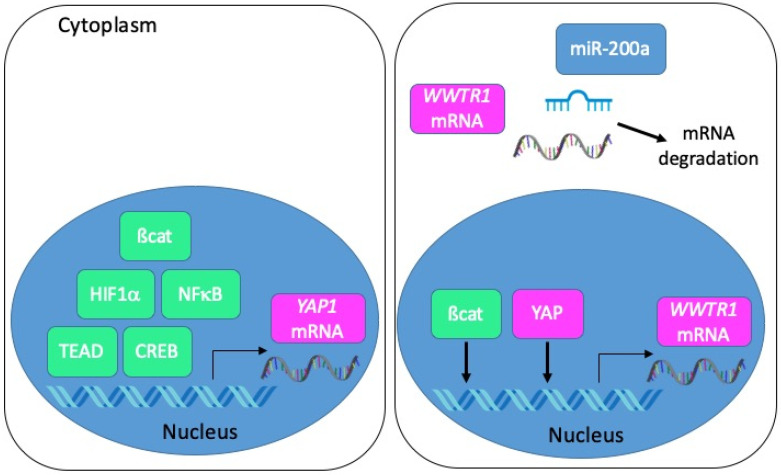
Mechanisms of YAP/TAZ transcriptional and post-transcriptional regulation in CRC. Left panel: the transcription of the *YAP1* gene is positively regulated by ß-catenin, HIF1α, NFκB, TEAD or CREB. Right panel: the transcription of the *WWTR1* gene is positively regulated by ß-catenin and YAP. Transcripts encoding TAZ are negatively regulated by miR-200a.

**Figure 6 cancers-12-03160-f006:**
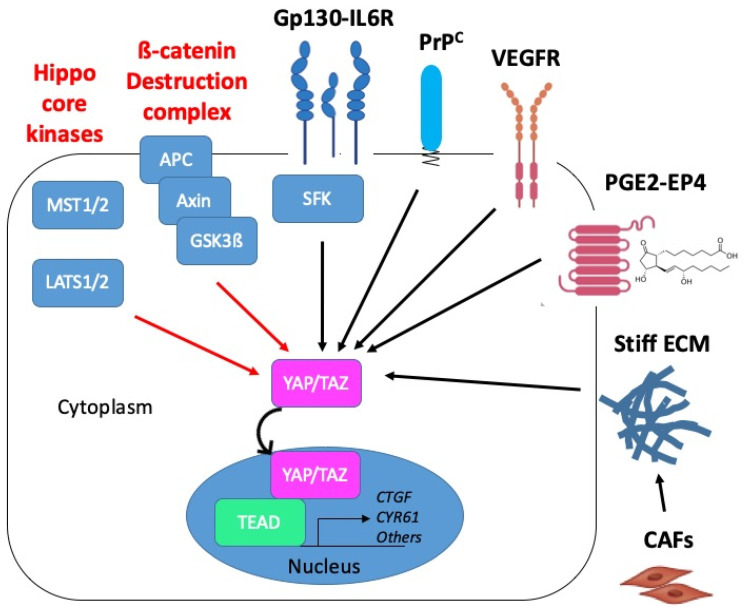
Proposed mechanisms of YAP/TAZ activation in CMS4 CRC. YAP/TAZ activation in CRC may operate through inactivation of negative regulatory mechanisms (red arrows): inactivation of the core kinases of the Hippo pathway, MST1/2 and LATS1/2, or inactivation of the ß-catenin destruction complex composed of APC, Axin and GSK3ß. YAP/TAZ activation can also occur through positive mechanisms of the regulation (black arrows) such as downstream from Gp130 via the src kinase family (SFK). In CMS4 CRC, YAP/TAZ activation is promoted by the cellular prion protein PrP^C^. Candidate upstream regulators of YAP/TAZ in CMS4 CRC are VEGFR, prostaglandins (PGE2) acting through the G-protein receptor coupled (GPCR) EP4 receptor or matrix stiffness resulting from the tumour infiltration by cancer-associated fibroblasts (CAF). Red and black arrows indicate negative and positive mechanisms of regulation, respectively.

**Figure 7 cancers-12-03160-f007:**
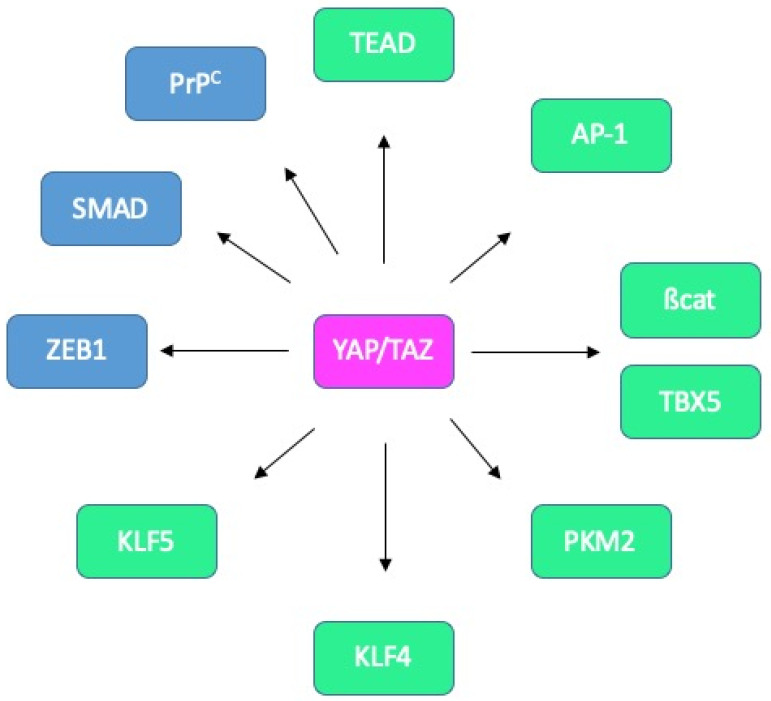
Known and potential binding partners of YAP/TAZ in CRC. Interactions between YAP/TAZ and TEAD, AP-1, ß-catenin/TBX5, PKM2, KLF4 and KLF5 (green) have been reported in CRC cells. Candidate partners in CRC cells (blue) include ZEB1, SMADs and PrP^C^.

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
