# Peer review of "YAP/TAZ Signalling in Colorectal Cancer: Lessons from Consensus Molecular Subtypes"

_cancers, 2020, doi:10.3390/cancers12113160_

Round 1
Reviewer 1 Report
In the manuscript, the authors reviewed possible involvement of YAP/TAZ in colorectal cancer. Colorectal cancer is a heterogeneous disease whose classification has undergone significant changes over the last few years. Based on their molecular biology characteristics and mutational and epigenetic status, human colorectal carcinomas were divided into four so-called consensus molecular subtype (CMS) groups. The possible role of YAP / TAZ within individual CMS groups of intestinal tumors is the focus of the review article. The article is well and clearly written. I also appreciate that the chosen topic (perspective) is very interesting from my point of view and “fills the gap” that exists in the literature. I have only a few minor comments:
(1) While reading, I was somewhat irritated by the inconsistent use of abbreviations. Here are just a few examples (please note that the list is probably not complete): (a) the abbreviation 'CRC' was introduced in the first sentence of Introduction and subsequently used, however in some parts of the text the full term (colorectal cancer) is used again and repeatedly; there is similar inconsistent usage of the abbreviation ‘ISCs’; (b) corresponding abbreviation is used first, but it is explained only in the following text (e.g. abbreviation ‘OS’); (c) the abbreviation is introduced, however the term is used only once in the whole main text (e.g. Hazard Ratio 'HR'; disease-free survival ‘DFS’; ‘SFK’; ‘AOM’); (d) the abbreviation is used but not explained (e.g. ‘TNM’); (e) the abbreviation was introduced but not used (e.g. ‘TCPA’); (f) Incorrect abbreviation ‘et al.’ (et al); inconsistent designation of the ApcMin strain (ApcMin versus ApcMin); (g) gene abbreviations should be written in italics.
(2) Information that TAZ is encoded by the WWTR1 gene is repeated several times in the main text.
Author Response
(a) the abbreviation 'CRC' was introduced in the first sentence of Introduction and subsequently used, however in some parts of the text the full term (colorectal cancer) is used again and repeatedly; there is similar inconsistent usage of the abbreviation ‘ISCs’;
The full term “colorectal cancer” has been replaced by CRC throughout the text. Likewise, the full-term “intestinal stem cells” has been replaced by ISCs
(b) corresponding abbreviation is used first, but it is explained only in the following text (e.g. abbreviation ‘OS’);
The abbreviation OS is now introduced when first used (page 2, lane 51 instead of lane 53)
(c) the abbreviation is introduced, however the term is used only once in the whole main text (e.g. Hazard Ratio 'HR'; disease-free survival ‘DFS’; ‘SFK’; ‘AOM’);
The corresponding abbreviations have been removed, except for SFK since is also features in Figure 6.
(d) the abbreviation is used but not explained (e.g. ‘TNM’);
The abbreviation TNM is now explained page 4.
(e) the abbreviation was introduced but not used (e.g. ‘TCPA’);
The abbreviation TCPA has been removed.
(f) Incorrect abbreviation ‘et al.’ (et al); inconsistent designation of the ApcMin strain (ApcMin versus ApcMin);
We have corrected “et al” into “et al.” and ApcMin into ApcMin.
(g) gene abbreviations should be written in italics.
Gene abbreviations are now written in italics. This has been corrected in the Figures et their legends as well.
(2) Information that TAZ is encoded by the WWTR1 gene is repeated several times in the main text.
We have removed this information when repeated.
Reviewer 2 Report
In this review, the authors tried to discuss the implication of YAP and TAZ in colon cancers with four distinct molecular subtypes. The subtypes need be considered to appropriately manage colon cancers. Therefore, this review is meaningful and should be published to trigger and solicit discussion and further studies in this line.
I have several suggestions, although I do not particularly request the authors to modify their manuscript according to my comments.
- It might be better to reconsider the size of each figure. Figure 5 is too big. I hope that the authors could show the figures in a more elegant style.
- The authors should be careful for the abbreviations. In line 45, they write “OS”, which is explained as “overall survival” in line 47. Figure 3B is labelled with “RPPA” but there is no explanation for RPPA as “Reverse Phase Protein Assay”. Readers will have difficulties to figure out the meanings of abbreviations in the text. Although it may not be consistent with the instructions of “Cancers”, the list of abbreviations is always helpful.
- They used “YAP/TAZ target score” in Figure 3. It would be more reader-friendly if they add the brief explanation for this score. Otherwise, readers must check the paper by Wang et al.
- In Section 3 and Figure 4, the authors should explain how they classified colon cancer cell lines as CMS1, 2, 3, and 4. Did they apply the CMS classifier, which was reported in the paper by Guinney et al., to the cell lines? It is desirable if they add the list of names of cell lines for each subtype.
- Is the subtitle of Section 2 “YAP/TAZ in colorectal cancer: from cells to patients” appropriate? In this section, the authors discussed “Clinical data” and then “Animal models”; from patients to animals. They did not discuss YAP and TAZ at the cell level.
- They reviewed the previously published papers using animal models and discussed the discrepancies. This part is interesting. I, who am not a specialist of colon cancers, would like to ask from curiosity about the possibility that four CMS subgroups have distinct cellular origins.
- They write in line 313 “it is tempting to speculate that YAP/TAZ activation in CMS4 tumors cells may also be driven by signals emanating form the tumour microenvironment such as PGE2.” They also cited their own work in line 295, supporting “In the context of CMS4 CRC, YAP/TAZ are under the control of PrPc.” Moreover, they describe in line 353 “Such a model is actually supported by the combined control by YAP/TAZ and TGFb on a panel of CMS4 genes downstream form PrPc.” Reading these lines, I speculate that the authors want to conclude that If YAP/TAZ are activated in some way and co-operate with the adequate partner, colon cancers gain CMS4 properties and that the high activation of YAP/TAZ is essential to maintain CMS4 phenotype. That is; No YAP/TAZ activation, no CMS4. Is this true? Is there no CMS4 colon cancers with the low activation of YAP/TAZ? Readers will want to know the authors’ frank opinion, even if speculative.
- In line 359, the authors state “further investigating YAP/TAZ through the lens of the CMS classification necessitates and extensive study of their potential binding partners in relation with CMS subtypes.” Reading this, I am a little bit confused. Do the authors also consider the possibility that YAP/TAZ play distinct roles in CMS1, 2, and 3, in which their expression is not so high? Is there any evidence that YAP/TAZ determine the properties of CMS1, 2, and 3? If this is the case, which roles can be attributed to YAP/TAZ in such colon cancers? Here again, readers will want to know the authors’ opinion.
Author Response
- It might be better to reconsider the size of each figure. Figure 5 is too big. I hope that the authors could show the figures in a more elegant style.
We agree with the referee that Figure 5 was too big. We have resized the Figure in this revised version.
- The authors should be careful for the abbreviations. In line 45, they write “OS”, which is explained as “overall survival” in line 47. Figure 3B is labelled with “RPPA” but there is no explanation for RPPA as “Reverse Phase Protein Assay”. Readers will have difficulties to figure out the meanings of abbreviations in the text. Although it may not be consistent with the instructions of “Cancers”, the list of abbreviations is always helpful.
The abbreviation OS is now detailed at first usage. The abbreviation RPPA is now explained.
- They used “YAP/TAZ target score” in Figure 3. It would be more reader-friendly if they add the brief explanation for this score. Otherwise, readers must check the paper by Wang et al.
As appropriately suggested by the referee, we now mention in the text (page 8): “In addition, as observed with clinical samples, CMS4 CRC cell lines display much higher levels of the YAP/TAZ target score (Figure 4), which corresponds to the integrated mRNA expression of a selection of 22 YAP/TAZ target genes and was shown to represent a robust index of YAP/TAZ activation (Wang et al., 2018) ».
- In Section 3 and Figure 4, the authors should explain how they classified colon cancer cell lines as CMS1, 2, 3, and 4. Did they apply the CMS classifier, which was reported in the paper by Guinney et al., to the cell lines? It is desirable if they add the list of names of cell lines for each subtype.
We agree with the referee that this part was succinct. We have now explained that the analyses were made by combining transcriptomic data from (Medico et al., 2015) and molecular classification of cell lines according to (Sveen et al., 2018) (page 8). We choose not to indicate the names of the cell lines as there is a large number of cell lines and the reader can easily find this information in the paper by Sveen et al. (Sveen et al., 2018).
- Is the subtitle of Section 2 “YAP/TAZ in colorectal cancer: from cells to patients” appropriate? In this section, the authors discussed “Clinical data” and then “Animal models”; from patients to animals. They did not discuss YAP and TAZ at the cell level.
We thank the referee for pointing this mistake. The subsection entitled “3. YAP/TAZ in colorectal cancer: the cellular scale” was intended to feature as a subsection of chapter 2. It has been corrected into “2.3. The cellular scale”.
- They reviewed the previously published papers using animal models and discussed the discrepancies. This part is interesting. I, who am not a specialist of colon cancers, would like to ask from curiosity about the possibility that four CMS subgroups have distinct cellular origins.
This is a very relevant question. This topic is discussed in the very interesting review by Fessler and Medema (Fessler and Medema, 2016), which we now refer to page 7.
- They write in line 313 “it is tempting to speculate that YAP/TAZ activation in CMS4 tumors cells may also be driven by signals emanating from the tumour microenvironment such as PGE2.” They also cited their own work in line 295, supporting “In the context of CMS4 CRC, YAP/TAZ are under the control of PrPc.” Moreover, they describe in line 353 “Such a model is actually supported by the combined control by YAP/TAZ and TGFb on a panel of CMS4 genes downstream form PrPc.” Reading these lines, I speculate that the authors want to conclude that If YAP/TAZ are activated in some way and co-operate with the adequate partner, colon cancers gain CMS4 properties and that the high activation of YAP/TAZ is essential to maintain CMS4 phenotype. That is; No YAP/TAZ activation, no CMS4. Is this true? Is there no CMS4 colon cancers with the low activation of YAP/TAZ? Readers will want to know the authors’ frank opinion, even if speculative.
We agree with that the hypothesis that “If YAP/TAZ are activated in some way and co-operate with the adequate partner, colon cancers gain CMS4 properties and that the high activation of YAP/TAZ is essential to maintain CMS4 phenotype” is worth considering, although there is no direct evidence until now to say this is true. We thank the referee for clearly asking the question and suggesting that we state our opinion, which we now do in page 12 at the end of section 3. We actually borrow the phrasing from the referee as it is very explicit and indeed corresponds to our view.
- In line 359, the authors state “further investigating YAP/TAZ through the lens of the CMS classification necessitates and extensive study of their potential binding partners in relation with CMS subtypes.” Reading this, I am a little bit confused. Do the authors also consider the possibility that YAP/TAZ play distinct roles in CMS1, 2, and 3, in which their expression is not so high? Is there any evidence that YAP/TAZ determine the properties of CMS1, 2, and 3? If this is the case, which roles can be attributed to YAP/TAZ in such colon cancers? Here again, readers will want to know the authors’ opinion.
To be honest, although we are convinced that YAP/TAZ play important roles in the progression of CMS4 CRC, we cannot exclude that they exert a different action in other CMS subtypes, where they are less abundant but still expressed. This issue has to be brought together with the controversial data obtained in mouse models. Of course, the key question to us is whether we can ascribe an indispensable for YAP/TAZ in CMS4 CRC. We have slightly rephrased this part to put the emphasis on CMS4.
Reviewer 3 Report
The manuscript reviews the accumulating evidence on YAP/TAZ in defining colorectal cancer subtypes. Overall, this is an exhaustive review article that is well written and largely documents the literature associated with the Hippo pathway effectors in regulating CRCs. I have a few comments to improve the readability and robustness of this article so as to make it more engaging for the readers.
1) A section describing the role of mechanical inputs in regulating YAP/TAZ in CRC needs to be included.
2) Figure 3: is it possible to generate a survival curve in these datasets based on YAP1 and or WWTR1 expression. This would depict the overall poor prognosis in CMS4 subtypes.
3) what are the levels of tumor angiogenesis amongst these 4 types of CRCs? This aspect needs a discussion in the context of YAP/TAZ expression. Given that YAP/TAZ controls angiogenesis (quote the references) and several upstream factors also regulate YAP in angiogenesis in CRCs and other cancers as well. This will also add a significance to the role of YAP/TAZ in CRCs along with the broader context of other cancers. Some key references in this regards to being added are PMID: 29535383; PMID: 31052445; PMID: 31052445; PMID: 27300434
Author Response
1) A section describing the role of mechanical inputs in regulating YAP/TAZ in CRC needs to be included.
We agree with the referee that this information was poorly mentioned in our initial version. We have added the following text in the section entitled “upstream regulators”: “Finally, other well-described upstream regulators of YAP/TAZ include various types of mechanical inputs (see (Ma et al., 2019; Panciera et al., 2017; Totaro et al., 2018) for review). Although not examined to our knowledge in the context of CRC, the mechanotransduction-dependent activation of YAP/TAZ was found to play an important role in intestinal repair after tissue damage (Yui et al., 2018) ». We choose not to elaborate further on the link between mechanical inputs and YAP/TAZ since this topic has not been addressed in the context of CRC to our knowledge. We refer the reader to the excellent reviews on the topics, which are now cited in this paragraph.
2) Figure 3: is it possible to generate a survival curve in these datasets based on YAP1 and or WWTR1 expression. This would depict the overall poor prognosis in CMS4 subtypes.
This is a very helpful suggestion. We have added a survival curve in the GSE39582 dataset based on the YAP/TAZ target score, since this analysis has not been reported previously, contrary to YAP1 and WWTR1. This analysis clearly show that high YAP/TAZ target scores are associated with poor prognosis.
3) what are the levels of tumor angiogenesis amongst these 4 types of CRCs? This aspect needs a discussion in the context of YAP/TAZ expression. Given that YAP/TAZ controls angiogenesis (quote the references) and several upstream factors also regulate YAP in angiogenesis in CRCs and other cancers as well. This will also add a significance to the role of YAP/TAZ in CRCs along with the broader context of other cancers. Some key references in this regards to being added are PMID: 29535383; PMID: 31052445; PMID: 31052445; PMID: 27300434
We thank the referee for this fruitful comment. CMS4 CRC is indeed associated with angiogenesis (which we now mention in the introduction section). We have inserted a new paragraph in the section entitled “upstream regulators” regarding the potential activation of YAP/TAZ in CMS4 CRC downstream form VEGF-VEGFR. Figure 6 has also been modified accordingly.
In a related aspect, we further added the notion that “YAP endows CRC cells with the ability to actively migrate within the vasculature and thereby enhance their metastatic spread (Benjamin et al., 2020) »